# Single-cell transcriptomic atlas of enteroendocrine cells along the murine gastrointestinal tract

**Christopher A. Smith**[1][�], **Elisabeth A. A. O'Flaherty**[1][�], **Nunzio Guccio**[1], **Austin Punnoose**[1], **Tamana Darwish**[1], **Jo E. Lewis**[1], **Rachel E. Foreman**[1¤a], **Joyce Li**[2], **Richard G. Kay**[1], **Alice E. Adriaenssens**[1¤b], **Frank Reimann**[1‡*], **Fiona M. Gribble**[1‡*]

**1** Institute of Metabolic Science, Metabolic Research Laboratories, Addenbrooke's Hospital, Cambridge, United Kingdom, **2** Department of Medicine, University of Massachusetts Medical School, Worcester, Massachusetts, United States of America

উ These authors contributed equally to this work.
¤a Current address: Integrated Bioanalysis, Clinical Pharmacology & Safety Sciences, AstraZeneca, Biomedical Campus, Cambridge, United Kingdom
¤b Current address: Division of Biosciences, University College London, London, United Kingdom
‡ FR and FMG are joint senior authors on this work.
\* fr222@cam.ac.uk (FR); fmg23@cam.ac.uk (FMG)

**Data Availability Statement:** Raw RNA sequencing data, count matrices, and Seurat objects per region are stored on the GEO repository (GSE 269778).

## Abstract

### Background

Enteroendocrine cells (EECs) produce over 20 gut hormones which contribute to intestinal physiology, nutrient metabolism and the regulation of food intake. The objective of this study was to generate a comprehensive transcriptomic map of mouse EECs from the stomach to the rectum.

### Methods

EECs were purified by flow-cytometry from the stomach, upper small intestine, lower small intestine, caecum and large intestine of NeuroD1-Cre mice, and analysed by single cell RNA sequencing. Regional datasets were analysed bioinformatically and combined into a large cluster map. Findings were validated by L-cell calcium imaging and measurements of CCK secretion in vitro.

### Results

20,006 EECs across the full gastrointestinal tract could be subdivided based on their full transcriptome into 10 major clusters, each exhibiting a different pattern of gut hormone expression. EECs from the stomach were largely distinct from those found more distally, even when expressing the same hormone. Cell clustering was also observed when performed only using genes related to GPCR cell signalling, revealing GPCRs predominating in different EEC populations. Mc4r was expressed in 55% of Cck-expressing cells in the upper small intestine, where MC4R agonism was found to stimulate CCK release in primary cultures. Many individual EECs expressed more than one hormone as well as machinery for

**Funding:** All funding for the study was provided from the following sources: Wellcome Trust (220271/Z/20/Z); Medical Research Council (MRC_MC_UU_12012/3); the MS instrument was funded by the MRC "Enhancing UK clinical research" grant (MR/M009041/1); MRL Genomics and Transcriptomics Core, Disease Model Core and Peptidomics Core, supported by MRC (MRC_MC_UU_12012/5); REF was funded by a BBSRC-iCASE studentship partnered with LGC. LGC provided support in the form of partial salary support for REF, but did not have any additional role in the study design, data collection and analysis, decision to publish, or preparation of the manuscript. The specific roles of this author are articulated in the 'author contributions' section. There was no additional external funding received for this study. After completing the study, REF moved to work at AstraZeneca. AstraZeneca had no involvement with this study.

**Competing interests:** "FMG and FR have received funding for other projects from AstraZeneca and Eli Lilly. LGC provided partial salary support for REF, but was not otherwise involved with this study. Since completing the work, REF moved to work for AstraZeneca which has had no involvement with the study. This does not alter our adherence to PLOS ONE policies on sharing data and materials."

activation by multiple nutrients, which was supported by the finding that the majority of L-cells exhibited calcium responses to multiple stimuli.

## Conclusions

This comprehensive transcriptomic map of mouse EECs reveals patterns of GPCR and hormone co-expression that should be helpful in predicting the effects of nutritional and pharmacological stimuli on EECs from different regions of the gut. The finding that MC4R agonism stimulates CCK secretion adds to our understanding of the melanocortin system.

## Introduction

Enteroendocrine cells (EECs) lie scattered in the epithelium of the gut from the stomach to the rectum and are the major intestinal chemosensors of nutritional and luminal stimuli. Their hormonal products play important roles in coordinating the function of the gastrointestinal (GI) tract and regulating metabolic processes such as insulin secretion and appetite. The best studied EECs are those producing glucagon-like peptide-1 (GLP-1) owing to the clinical success of injectable GLP-1 receptor agonists for the treatment of type 2 diabetes and obesity [1]. However, the gut produces over 20 different bioactive peptides, some from distinct cell populations and others from overlapping cell types [2, 3]. Understanding the release of different gut peptides is of increasing translational interest due to the clinical efficacy of bariatric surgery which increases postprandial release of multiple gut hormones [4] and the emerging effectiveness of drugs targeting multiple gut hormone receptors for obesity and diabetes [1].

The traditional naming of EECs is based on a one-cell-type one-hormone (or occasionally two hormone) system, in which D-cells produce somatostatin (SST), G-cells produce gastrin, I-cells produce cholecystokinin (CCK), K-cells produce glucose-dependent insulinotropic polypeptide (GIP), L-cells produce GLP-1 and peptide YY (PYY), N-cells produce neurotensin (NTS), S-cells produce secretin (SCT), X-cells produce ghrelin, enterochromaffin (EC) cells produce serotonin and enterochromaffin-like (ECL) cells produce histamine. However, this nomenclature is based on staining characteristics which typically only detect the most highly abundant hormones in a cell [3]. Transcriptomic analysis of purified EEC populations revealed a more subtle picture with more overlap between cell types than previously believed [5, 6]. Single cell transcriptomic analyses of EECs from defined intestinal regions refined this idea, reporting that D-cells, EC cells and K-cells usually form distinct cell clusters, whereas I-, L-, N- and S-cells form an overlapping cell group [7, 8]. Whilst such single cell cluster maps have been generated for EECs from the small intestine [9–11] and colon [12], a comprehensive map of EECs along the full length of the gut, allowing comparisons of EECs across different regions, is not yet available. Transcriptomic analyses of EECs in mice and humans have repeatedly identified the same set of nutrient-sensing G-protein coupled receptors (GPCR), regardless of the EEC sub-type or their location. These include receptors for long chain fatty acids (*Ffar1*, *Ffar4*), short chain fatty acids (*Ffar2*, *Ffar3*), monoacylglyerols (*Gpr119*), aromatic amino acids (*Casr*, *Gpr142*) and bile acids (*Gpbar1*) [13].

As EECs comprise only approximately 1% of the intestinal epithelium, and an even lower percentage of cells in a full thickness gut biopsy, they are often sparsely represented in single cell atlases that do not include an EEC enrichment purification step. In this study, we performed a comprehensive single cell RNA sequencing (scRNA-seq) analysis of EECs from the stomach through to the rectum of mice, using a NeuroD1-dependent fluorescent marker to

purify EECs prior to sequencing. As well as identifying cell clusters representing the major hormonal secreted products, this enabled comparisons of EECs from different regions and separated EEC populations by their dominant GPCR signalling pathways. An example of identified differential GPCR expression was *Mc4r* in *Cck*-expressing cells, the functional relevance of which was validated by showing that MC4R agonism stimulated CCK secretion from intestinal cultures.

## Methods

### Animal work and ethics

All animal procedures were carried out in accordance with the Animals (Scientific Procedures) Act 1986 Amendment Regulations 2012, following ethical review by the University of Cambridge Animal Welfare and Ethical Review Body (AWERB). The animal work was performed under UK Home Office Project License PE5OF6065. Mice were group housed with ad libitum access to regular chow and water. NeuroD1-cre mice [14] were crossed with ROSA26-EYFP mice to generate NeuroD1-cre x EYFP mice which expressed EYFP under the pan-EEC marker NeuroD1. Mice were maintained under a 12-hour light/dark cycle (lights on 07:00 to 19:00 GMT) in individually ventilated cages in temperature- and humidity-controlled holding rooms.

### Single-cell sample preparation and solutions

Adult NeuroD1-Cre x EYFP mice were culled by cervical dislocation and GI tracts were removed. To achieve sufficient cell numbers for scRNA-seq, four males (stomach); one female (USI); five females (LSI); four females (caecum) and two females (LI) were used. Prior to cell sorting, each single cell preparation was transferred to FACS solution: HBSS (no $Ca^{2+}$ or $Mg^{2+}$), 10 µM Y-27632, 10% FBS, 1:1000 DAPI, and 1:1000 DRAQ5.

Stomachs were digested as described previously [15]. Caecum was digested as in [12]. For the small intestine, 10 cm lengths of intestine distal to the stomach (USI) and 10 cm proximal to the caecum (LSI) were excised and processed separately. Mesentery and exterior smooth muscle layer were removed and discarded, intestinal sections were cut into 1–2 cm segments and rinsed in PBS. Segments were transferred to solution A (PBS (no $Ca^{2+}$ or $Mg^{2+}$), 15 mM EDTA, 1 mM DTT) and incubated for 7 min at room temperature (RT). They were then transferred to a tube containing ice cold solution B (PBS (no $Ca^{2+}$ or $Mg^{2+}$), 10 µM Y-27632), and shaken gently, before being placed back in solution A. Transfers from solution A to B were repeated for 5 incubation periods (35 min total). Tissue was centrifuged at 300 g for 5 min at 4˚C, then resuspended in 10 ml digestion media (Trypsin 0.25% EDTA, 0.1 mg/ml DNAse1) and incubated for 5 min at 37˚C, shaking gently at the beginning and end of the incubation. Supernatant was removed and centrifuged at 500 g for 5 min at 4˚C to collect all single cells. The resulting pellet was triturated in washing media (HBSS (no $Ca^{2+}$ or $Mg^{2+}$), 10% FBS, 5 µM Y-27632) and centrifuged at 500 g for 5 min at 4˚C. The pellet was resuspended in washing media, passed through 100 µm strainer and centrifuged at 500 g for 5 min at 4˚C. The pellet was resuspended in washing media and passed through 50 µm strainer and centrifuged at 300 g for 5 min at 4˚C. The pellet was resuspended in FACS solution and kept on ice.

Single cell suspensions from each GI region were sorted using an Influx Cell Sorter (BD Biosciences, Franklin Lakes, USA) at the Cambridge Institute of Medical Research (CIMR) Flow Cytometry Core Facility. DAPI and DRAQ5 staining, side scatter, forward scatter and pulse width gates were used to remove debris and cell clusters. Single cell preparations resulted in FACS purified samples of 16,000 cells from each the stomach, LSI and caecum, 12,000 cells

from the USI and 7,000 cells from the LI, which were collected in FACS solution and transferred to genomics core facilities on ice.

## Single-cell RNA-sequencing

cDNA library preparations were performed using the Chromium system (10X Genomics, Pleasanton, USA) by the Genomics Core Facilities at the Cancer Research UK (CRUK) Cambridge Institute (for the USI, LSI and LI) and the Wellcome-MRC Cambridge Stem Cell Institute (for the stomach and caecum). Libraries for the LI were sequenced on the HiSeq 4000 system (Illumina, San Diego, USA) while all other GI regions were sequenced on the NovaSeq 6000 system (Illumina).

Quality controls, read alignment (with reference to the mm39 genome downloaded from the UCSC genome browser) and raw count quantification for each cell were generated using the CellRanger pipeline (v6.1.2, 10x Genomics). Sequencing data previously published for the LI [12] were re-aligned to the updated genome for direct comparison with the other datasets.

Analyses from raw counts were performed using the Seurat package (v4.4.0) in R [16]. Samples were initially filtered per region so that each gene be detected in at least 3 cells, and each gene have at least 500 (USI and LSI) or 1000 (stomach, caecum and LI) raw counts across all samples per region. Samples were further filtered so that the percentage of mitochondria-encoded genes (%mito) be less than 20% (USI, LSI and LI) or 25% (stomach and caecum) of all raw counts per sample. Of the filtered data, the median (min–max) number of reads per cell were 35,679 (2,140–389,084) in the stomach, 15,440 (718–261,381) in the USI, 6,462 (650–216,128) in the LSI, 5,664 (1,648–104,978) in the caecum, and 4,438 (1,402–40,537) in the LI dataset. The median (min–max) number of genes per cell were 3,276 (1,002–9,330) in the stomach, 3,663 (504–9,350) in the USI, 2,487 (501–9,367) in the LSI, 2,043 (1,001–8,308) in the caecum, and 1,943 (1,001–5,935) in the LI dataset. Raw count data were normalised using SCTransform, regressing %mito. Principal component analysis (PCA) was performed, and the first n PCs used to inform downstream t-distributed Stochastic Neighbour Embedding (tSNE) and clustering analyses: n = 8 for stomach, 11 for USI, 20 for LSI, 10 for caecum, 11 for LI. Clustering of cells was performed by k-means, with k = 10 and resolution = 1.0 (USI, LSI and LI) or 1.5 (stomach and caecum). Differential expression analysis was calculated using negative binomial regression method, calculating only positively enriched genes per cluster, and minimum percentage 25% of cells within the cluster having counts > 0 (min.pct = 0.25). Genes were defined as being positively expressed in a sample if their raw count was greater than that gene's threshold per region. The threshold was calculated using Huang thresholding [17] (Huang2 method in autothresholdr's auto_thresh function) on the distribution of counts per million (CPM) per gene after combining cell data from all regions–thresholds were calculated separately for *Sst* and *Ghrl* in cells from the stomach to account for the high background expression of each gene in this region.

Datasets derived from individual regions were integrated using Seurat's canonical correlation analysis (CCA) driven integration of SCTransform-normalised datasets. The list of variable genes for integration was defined as the union of all variable genes per dataset, and the number of integration features used was 2000.

G-protein coupled receptor (GPCR) cell signalling space was defined as the list of genes for GPCRs, G-protein subunits and their regulators, GPCR kinases, adenylate cyclases, phosphodiesterases, beta-arrestins, isoforms of phospholipase C and protein kinase A and C, protein kinase anchoring proteins, ion channels, mediators of calcium release, and ryanodine receptors (full list in supporting information). The tSNE plot in GPCR signalling space was

calculated on the integrated dataset using the first 10 PCs after running PCA on the list of genes defined above.

## Integration with published datasets

Datasets derived from individual regions were integrated using Seurat's canonical correlation analysis (CCA) driven integration of SCTransform-normalised datasets. The list of variable genes for integration was defined as the union of all variable genes per dataset, and the number of integration features used was 2000. Upper and lower small intestine datasets from this publication were integrated with previously-published mouse small intestine epithelial cell data [7] extracted from the Gene Expression Omnibus (GEO; GSE92332), whereas the large intestine dataset from this publication was integrated with previously-published mouse colon cell data [18] extracted from the GEO (GSE245188). Data from published datasets were normalised using SCTransform within Seurat prior to integration. Integration was performed as above.

## Hormone secretion from primary mouse tissue

Saline buffer used for hormone secretion and calcium imaging experiments contained 138 mM NaCl, 4.5 mM KCl, 4.2 mM NaHCO$_3$, 1.2 mM NaH$_2$PO$_4$, 2.6 mM CaCl$_2$, 1.2 mM MgCl$_2$, 10 mM HEPES; adjusted to pH 7.4 with NaOH.

Primary cultures of upper small intestine (10cm lengths distal to the stomach) were prepared as described previously [19]. Crypt suspensions were distributed into wells of a 24-well plate pre-treated with 2% Basement Matrix Extract (R&D Systems, #3433), and incubated at room temperature for 15 min before incubating at 37˚C and 5% CO$_2$ overnight. Adhered cells were washed 3x for 10 min at 37˚C using fresh saline buffer containing 1 mM glucose (Sigma-Aldrich G7528) and 0.001% BSA (Fraction V, Scientific Laboratory Supplies) for LC-MS analysis or 0.1% BSA for immunoassay. Cells were stimulated for 2 hr at 37˚C in saline buffer with the appropriate BSA concentration, 10 mM glucose, 2% DMSO (vehicle) and individual test compounds: 10 µM 3-Isobutyl-1-methylxanthine (IBMX; Sigma-Aldrich I7018), 10 µM forskolin (Sigma-Aldrich F6886); 1 µM setmelanotide (Cayman Chemical 35564). Stock solutions of IBMX and forskolin were prepared in DMSO, and setmelanotide in sterile-filtered MilliQ H$_2$O. After incubation, stimulation buffer was removed and centrifuged at 2,000 g for 5 min, snap-frozen on dry ice, and stored at -80˚C prior to either mass spectrometry or immunoassay. ELISA for measurement of total GIP was performed as per manufacturer's instructions (Millipore, EZRMGIP-55K).

## Liquid chromatography-mass spectrometry

The mouse preproCCK(21–44) assay was adapted from our previously-described protocol to measure human preproCCK(21–44) [20]. A calibration line of mouse preproCCK(21–44) (1–1000 pg/ml; Cambridge Research Biochemicals, Billingham, UK) was prepared in 0.001% BSA in water, and extracted alongside supernatant samples. Supernatants were stored at -80˚C prior to extraction and thawed on ice. 50 µl of internal standard solution (0.5 ng/ml of stable isotope labelled mouse preproCCK(21–44) (Cambridge Research Biochemicals) in 1% FA (formic acid) (aq.) was spiked into 200 µl of the supernatant and standards in a Protein LoBind plate (Eppendorf). Samples were loaded onto an Oasis HLB PRiME µElution SPE plate (Waters) and washed with 200 µl of 0.1% FA (aq.) and 200 µl of 5% methanol 1% acetic acid (aq.). The peptides were eluted with 2 x 30 µl of 60% methanol 5% acetic acid (aq.) into a Quan Recovery Plate (Waters) and diluted with 75 µl of 0.1% FA (aq.) prior to analysis. The extracted samples were analysed on a Thermo Fisher Ultimate 3000 nano-LC system coupled to a Q-Exactive Plus Orbitrap mass spectrometer in positive ion mode. A parallel reaction

monitoring method was used to target the $[M+3H]^{3+}$ charge state for the unlabelled and labelled preproCCK(21–44) peptides (890.46 m/z and 894.46 m/z respectively). Mouse preproCCK(21–44) was quantified in all samples using the Quan Browser software (Thermo Scientific Xcalibur).

## Organoids from mouse ileum

Mouse small intestine organoid media (ENR) contained 7.5–10% RSPO1 conditioned media, murine 50 ng/mL EGF (#PMG8043, Invitrogen, Waltham, USA), 1% penicillin/streptomycin (P/S) (100 units/mL), 100 ng/mL murine noggin (#250–38, PeproTech, London, UK), 2 mM L-Glutamine, 1X N2 supplement (#17502001, Invitrogen), 2X B27 supplement without vitamin A (#12587001, Invitrogen), 1 μM N-acetyl-L-cysteine and 10 μM Y-27632 (#1254/50, Tocris, Bristol, UK), prepared in ADF (Advanced DMEM/F-12) (#12634028, Gibco, Waltham, USA).

Organoids were generated from the GI tracts of GLU-Venus mice as previously described [7]. Crypt pellets following tissue processing were resuspended in 750 μl Cultrex® BME (basement membrane extract) (R&D Systems #3533), plated in 25 μl domes in a pre-warmed plate, and incubated at 37˚C for 30 min. ENR medium was added once the BME domes had polymerised. Fresh ENR media was added to cultures every 2–3 days, and organoids were passaged every 10–14 days.

## Calcium imaging of mouse ileum organoids

Live single-cell calcium imaging of Venus-expressing ileal murine L cells was performed as previously described [21]. Briefly, differentiated organoids were enzymatically and chemically dissociated into single cells or small cell clusters, seeded on 35 mm glass-bottom dishes (MatTek) that had been previously precoated with 2–4% Basement Matrix Extract (R&D Systems, #3433) and incubated overnight. At time of imaging, the cultures were washed with saline buffer, loaded with 5 μM intracellular calcium indicator fura2-AM (Thermo Fisher Scientific) and mounted on a 40X oil-objective on a fluorescence inverted microscope (Olympus IX71). Cells were excited at 340nm and 380nm every 2 s using Metafluor software (Molecular Devices) while cells were continuously perfused with saline buffer, in the presence and the absence of test reagents. Ratios of fura2-AM emission at 340 v 380 nm excitation were calculated following background subtraction. Responses are presented as fold change between the mean fura2 ratio over 10 time points (20 s) centred at the maximum value for fura2 ratio recorded during perfusion of stimulus, and the mean fura2 ratio over the 10 time points immediately prior to onset of stimulus. If the maximum value during perfusion occurred within 5 time points of the onset of stimulus, the response was calculated using the first 10 time points during perfusion. Cells whose 10 time points about the maximum were significantly increased relative to the 10 time points immediately prior to onset of stimulus (by Student's t-test, $p < 0.01$) were counted as responders.

## Results

EECs from the stomach, upper small intestine (USI), lower small intestine (LSI) and caecum of NeuroD1-Cre x Rosa26-EYFP reporter mice were flow-sorted and analysed by single cell RNA-seq using the 10xGenomics Chromium platform. ScRNA-seq data we previously published [12] from the mouse large intestine (LI) were reanalysed and included for comparison. Data from individual gut regions were analysed using the Seurat R package and plotted using t-distributed stochastic neighbour embedding (tSNE), revealing cell clusters which differed in their expression of known secretory peptides. Some non-EECs survived the cell sorting,

including proliferating cells and enterocytes, and were removed from subsequent analysis. In total, 20,006 EECs were included in the analysis (6,076 stomach, 4,787 USI, 4,157 LSI, 3,438 caecum and 1,548 LI). Cluster analysis identified a total of 10 EEC subgroups across all regions analysed (Fig 1).

## EEC subgroups in different intestinal regions

The numbers of cells in each cluster from different regions of the mouse gut are given in S1 Fig. Some EEC subgroups were detected in every region, including *Sst*-expressing D-cells, *Gcg*-expressing L-cells, and EC-cells labelled based on expression of tryptophan hydroxylase-1 (*Tph1* –a key enzyme for serotonin biosynthesis; Fig 1A). D-cells were most abundant in the stomach, where they made up 34% (n = 2,087) of all EECs, compared with 7–11% in the small and large intestine. By contrast, the proportion of L-cells increased from 3% in the USI to 35% in the LI. A cluster of cells in the LI had lower expression of *Gcg* compared with the main L-cell cluster, but expressed comparatively higher levels of *Insl5*, and were labelled as Insl5 cells (7%, n = 105). Interestingly, there were a small number of *Gcg*-expressing cells in the stomach (<1%, n = 29), consistent with previous reports of a population of gastric EECs producing glucagon in the mouse [2]. EC-cells were found throughout the entire gut, with the lowest percentage in the stomach (3%), compared with 36–82% in the small and large intestines and caecum.

Other EEC subgroups were found only in specific gut regions. *Ghrl*-expressing X-cells were mainly detected in the stomach (58%), with a few also in the USI (4%). The stomach was also the only region containing G-cell and ECL-cell clusters. Distinct *Gip*-expressing K-cell clusters were detected only in the small intestine, making up 22% of EECs in the USI and 14% in the LSI. Cell clusters characterised by high levels of *Cck*, labelled as I-cells, were specific to the USI, where they made up 24% of all EECs. *Nts*-expressing N-cells were found in every region of the gut except the stomach and large intestine and were most abundant in the LSI (24%).

We combined the data from our USI and LSI NeuroD1-labelled EECs with published unbiassed datasets [7, 18]. EECs from the current study showed good overlap with EEC populations in the unbiassed datasets (S1 Fig), and no additional EEC populations were evident in the full epithelial data that had not been picked up using the NeuroD1-Cre approach.

## Differential expression analysis of cell types and regions

To understand which genes best define each EEC subgroup, differential expression (DE) analysis was performed across all EEC subgroups and regions. Fig 1B depicts the top DE ($p_{adj} <$ 0.05, $\log_2$(fold-change) > 1) genes per cluster and per region, selecting for highest average expression, highest fold change compared with other cell types in the same region, and variability across similar cell types between regions. Genes enriched in the clusters labelled as L-cells include *Gcg*, *Sct*, *Cck*, *Pyy*, *Nts*, and *Insl5*.

Examining the expression of each of these genes by region revealed that although stomach contained very few *Gcg*-expressing cells, this cell population exhibited higher *Gcg* expression than L-cells elsewhere in the gut. Expression levels of *Cck* and *Sct* were highest in cell clusters from the small intestine with decreasing levels beyond the USI, whereas *Pyy* levels were higher in clusters further down the gut. In line with previous findings [11, 12], *Insl5* was most enriched in EECs from the LI. Pancreatic polypeptide, *Ppy*, was enriched in several cell clusters from the stomach and large intestine.

We specifically examined the expression of the prohormone convertases, *Pcsk1* and *Pcsk2*, as they underlie differential processing of prohormones such as proglucagon in different tissues (with PCSK1 typically generating GLP-1 in the gut and CNS, and PCSK2 producing

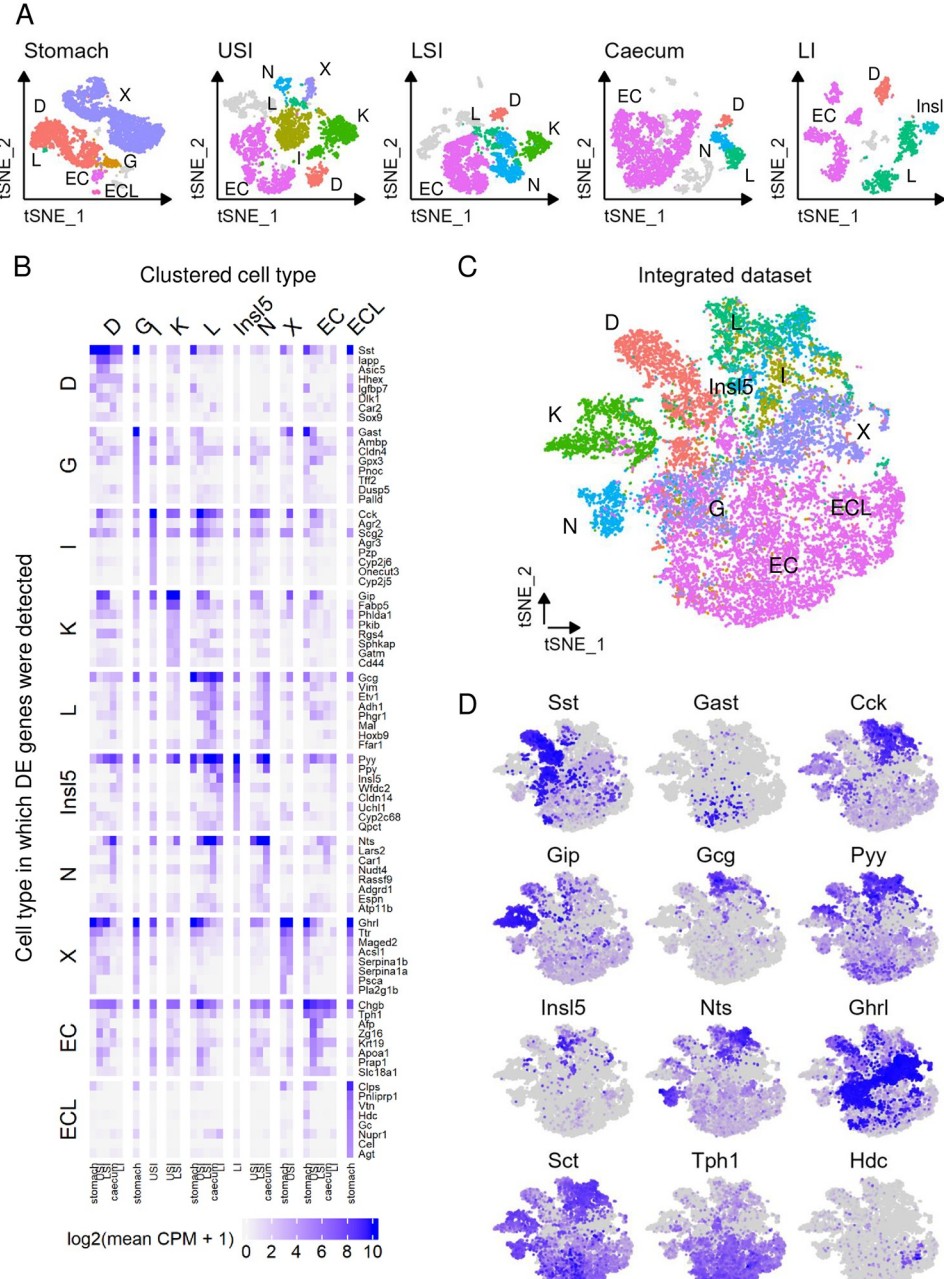

**Fig 1. Transcriptomic analysis of mouse EECs along the entire GI tract.** (A) t-distributed stochastic neighbour embedding (tSNE) plots of single cell RNA-sequencing (scRNA-seq) data of NeuroD1-positive cells from mouse (top to bottom) stomach, upper small intestine (USI), lower SI (LSI), caecum and large intestine (LI). Clusters labelled by standard enteroendocrine cell (EEC) nomenclature defined by expression of known gene markers. Non-EECs are shown in grey. (B) Heatmap of expression for top 10 differentially-expressed genes per EEC cluster, calculated per GI region. Expression is $\log_2$(mean counts per million + 1). Differentially-expressed genes defined as $p_{adj} < 0.05$, genes filtered for $\log_2$(fold change) > 1.5. (C) tSNE plot for integrated dataset of all EECs from all regions in (A). Samples labelled by EEC defined by clustering in individual datasets. (D) Marker expression per EEC in tSNE coordinates from (C). Expression is $\log_2$(mean counts per million + 1).

glucagon in the pancreas). Whereas *Pcsk1* expression was evident in all EECs, *Pcsk2* was most abundant in EECs from the stomach, particularly *Gcg*-expressing cells, D-cells and G-cells (S1 Fig). This is consistent with previous reports of a small population of glucagon-producing cells in the stomach of dogs [22] and mice [2], and of differential processing of prosomatostatin to generate SST-14 in the stomach compared with SST-28 in the intestine [2].

## Integration of datasets–overlapping cell types

To enable comparisons between EECs from different regions, all datasets were combined using canonical correlation analysis (CCA) within Seurat's functions for dataset integration (see methods). A map was generated including all EECs from all regions, plotted using tSNE coordinates. Most EEC subgroups formed distinct clusters regardless of the region from which they originated (Fig 1C and S1 Fig). This is likely driven in part by the expression of their principal secretory hormones, which showed distinct localisation on the cluster map (Fig 1D). By contrast, N-cells from the caecum, upper and lower SI separated away from each other, and stomach-derived X-cells were located separately from their small intestinal counterparts, although in a similar region of the map, suggesting that X-cells from the stomach are transcriptomically distinct compared with those from the intestine.

## Co-expression of secretory hormones

We next examined expression of hormonal markers on a per-cell basis rather than by cluster. A threshold in gene expression for each principal hormone was calculated using Huang thresholding, and any cell expressing greater than the calculated threshold was considered to express that marker (S2 Fig). Numbers of EECs per region expressing each hormone together with the corresponding gene expression level for that hormone in the positive cells are depicted in Fig 2A. Also shown in this figure are the percentages of cells co-expressing a second hormone (or not) and expression levels of the second hormone. Venn diagrams in Fig 2B show cell numbers co-expressing common hormonal combinations in different intestinal regions. Across the full dataset, the hormonal markers least likely to be co-expressed with a second hormone were *Sst*, *Ghrl*, *Tph1* and *Gip*, consistent with the separate clustering of D-, X-, EC- and K-cells. Low levels of *Hdc* expression were unexpectedly identified in some EECs in the distal intestine, predominantly colocalised in EC cells in the caecum and with L-cell hormones in the large intestine. Use of the Huang thresholding method identified broadly similar numbers of EECs as the clustering method.

In the USI, 93% of *Gcg*-expressing cells and 30% of *Nts*-expressing cells co-expressed *Cck*, but the majority of *Cck*-positive cells expressed neither *Gcg* nor *Nts* and instead had *Sct* as a major second hormonal product. Although the USI dataset did not have a distinct *Cck-Tph1* double-positive cluster as described by other groups [10, 11], a small proportion of cells did co-express both *Cck* and *Tph1*. In the LSI, *Sct* was found to be co-expressed with multiple other hormones, including *Cck*, *Nts*, *Pyy* and *Gip*–most notably 90% of *Pyy*-expressing cells also expressed *Sct*. 43% of EECs in the large intestine expressed at least one of *Gcg*, *Insl5* or *Pyy*, and very few cells (4%) expressed *Insl5* without *Pyy*. *Gcg* expression was only identified in a proportion of *Pyy*-positive cells (e.g ~50% in colon), even though immunostaining studies have estimated much higher proportions of cells containing both PYY and GLP-1 [5]; likely these *Pyy*-positive *Gcg*-negative cells at the transcript level reflect L-cell maturation, and that *Gcg* was expressed earlier in their lifespan, as suggested in previous studies [23, 24].

## GPCR cell signalling space

We next investigated whether different EEC subgroups could be separated solely by their expression of genes involved in GPCR-associated cell signalling pathways. To address this

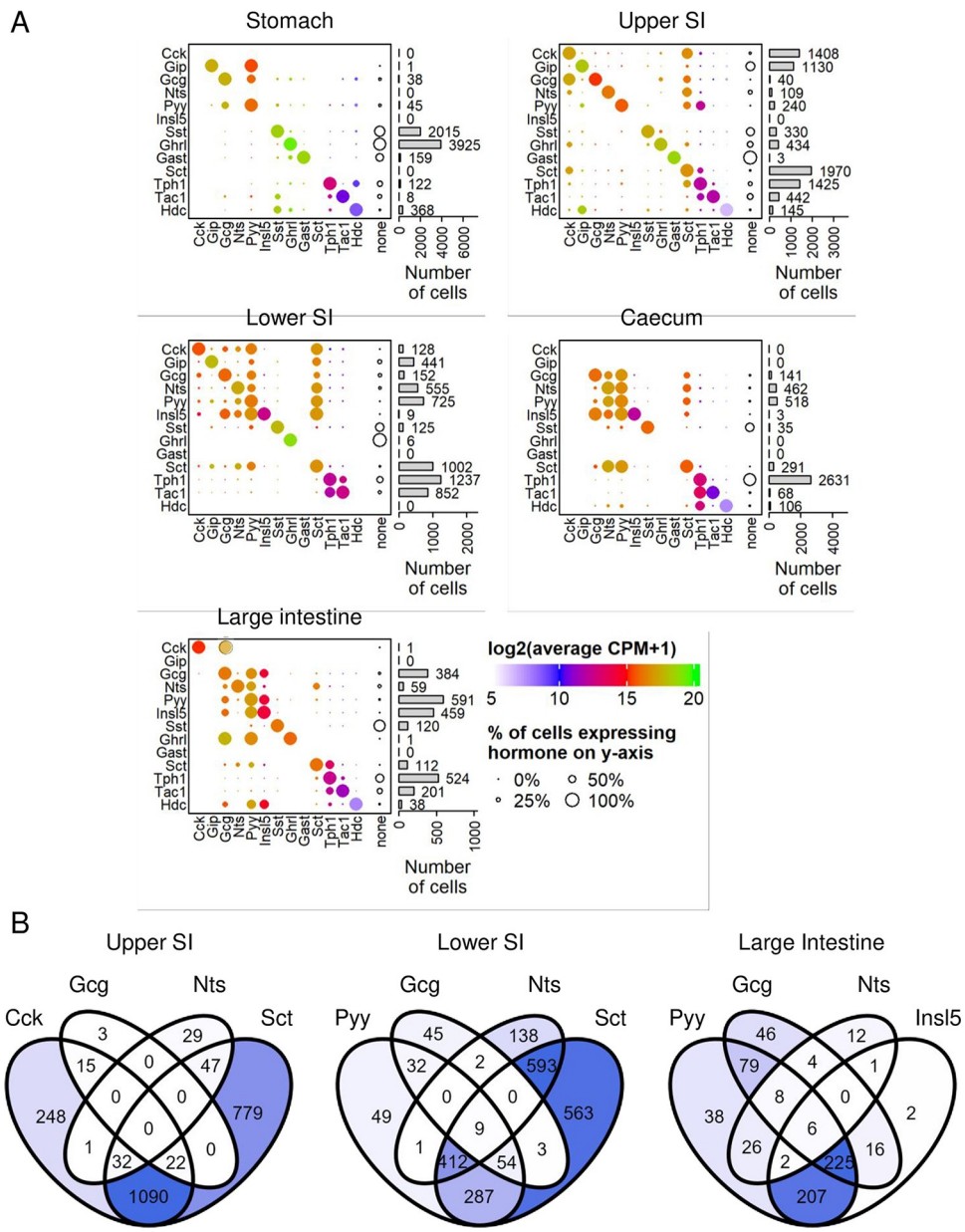

**Fig 2. Co-expression of hormonal markers.** (A) Dot plots representing co-expression of hormones or cell markers (*Tph1*, *Hdc*) per GI region. Dot size represents the percentage of cells expressing the gene on the y-axis, dot colour the expression of the gene on the x-axis in $\log_2$(mean counts per million + 1). Open circles labelled "none" represent percentages of cells expressing none of these additional hormonal markers. Bars on the right represent the number of cells expressing the hormone on the y-axis. (B) Venn diagrams demonstrating number of EECs co-expressing: top, *Cck*, *Gcg*, *Nts* and *Sct* in the upper SI; middle, *Sct*, *Gcg*, *Pyy* and *Nts* in the lower SI; bottom, *Insl5*, *Gcg*, *Nts* and *Pyy* in the LI. Darker blue represents larger cell number.

question, we rederived tSNE maps in a coordinate system defined only by GPCR signalling pathway and ion channel genes, in what we call the GPCR cell signalling space (Fig 3). Rather than calculating tSNE coordinates using genes calculated to be most variable within the data-set, only cell signalling-related genes were provided as an input for the clustering. This gene list included GPCRs, G-protein subunits and their regulators, GPCR kinases, adenylate cyclases, phosphodiesterases, beta-arrestins, isoforms of phospholipase C and protein kinase A

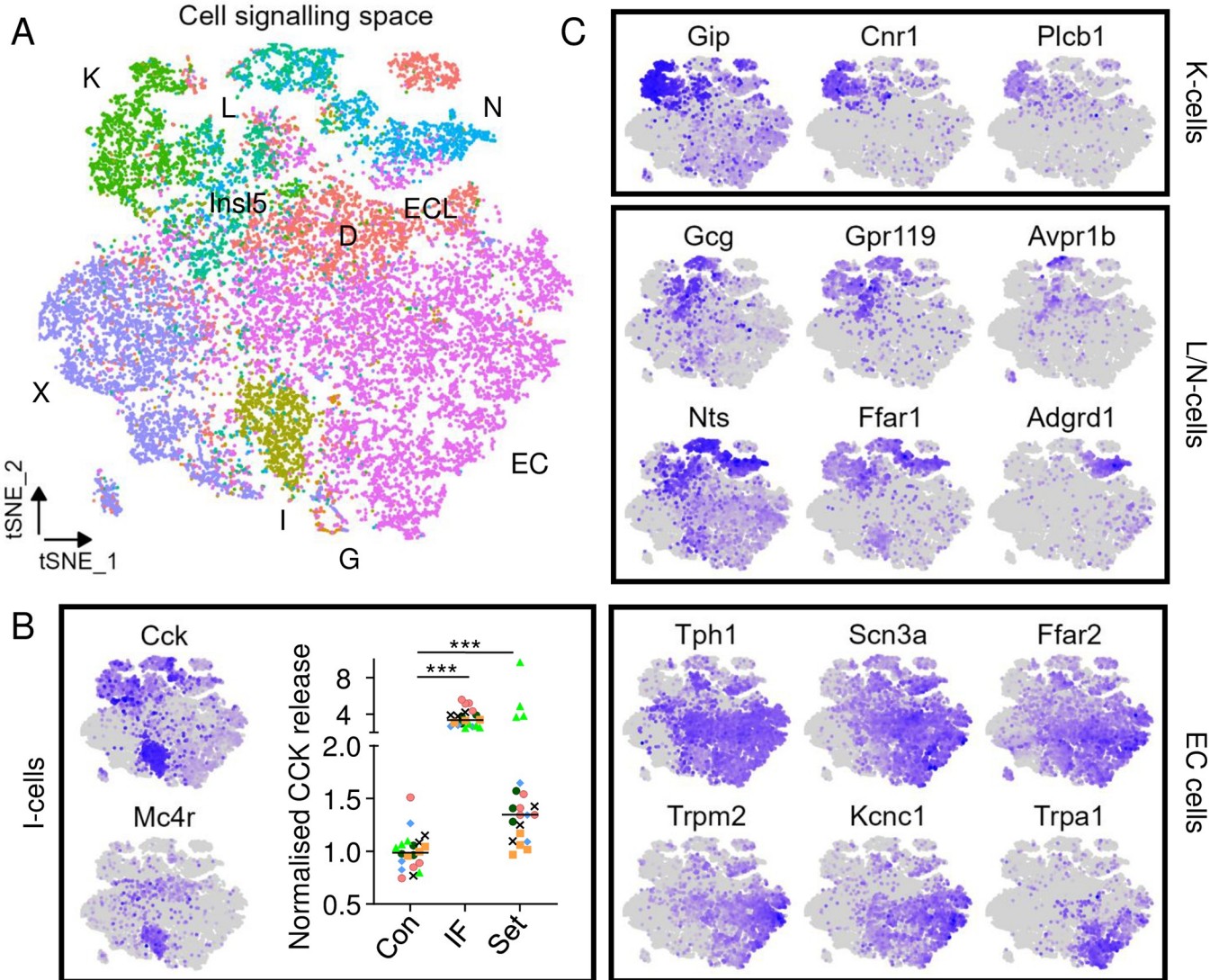

**Fig 3. EECs show sub-clustering based on GPCR cell signalling genes alone.** (A) tSNE plot for integrated dataset generated using genes involved in GPCR cell signalling. Samples labelled by EEC defined by clustering in individual datasets. (B) Left: Relative expression of *Cck* and *Mc4r*, which were each differentially-enriched ($p_{adj}$ < 0.05) in the I-cell-labelled cluster. Right: CCK secretion, measured as proCCK(21–44) by liquid-chromatography mass spectrometry, from mouse USI primary tissue after 2 hr stimulation with control (Con; 2% DMSO), IBMX (10 μM)/forskolin (10 μM) (IF), or setmelanotide (Set; 10 μM). Concentrations were normalised to mean concentrations in control wells from the same experiment. Horizontal lines represent mean concentration (n = 6 experiments, 3–4 samples per experiment, colour-coded by experiment). Statistical analysis by two-way ANOVA with experimental run as the second factor, and Dunnett's post-hoc test (*** p < 0.001). (C) Relative expression of selected differentially-expressed ($p_{adj}$ < 0.05) genes based on clustering analysis. Panels represent, from top to bottom, K-cells, L- and N-cells, and ECs.

and C, protein kinase anchoring proteins, ion channels, mediators of calcium release, and ryanodine receptors.

Mapping the full mouse EEC dataset in the GPCR cell signalling space interestingly revealed that cells continued to cluster according to their original EEC subgroup (as defined for each cell by their location in the full gene clustering model in Fig 1). This suggests that different EEC types have distinct cell signalling signatures, and therefore potentially different functional responsiveness to GPCR stimuli. Notably, whereas I-, L- and caecal N-cells overlapped in the original map (Fig 1), I-cells mapped separately from L- and N-cells in the GPCR cell signalling

space. X-cells again separated into 2 clusters determined by their stomach vs USI origin, suggesting that ghrelin secretion is differentially regulated in the stomach and small intestine.

Genes enriched in different areas of the GPCR cell signalling space map were calculated by performing cluster analysis and subsequent DE analysis per cluster (negative binomial model, $p_{adj} < 0.05$; Fig 3B and 3C). Notable DE genes across clusters include: melanocortin receptor *Mc4r* which was enriched in I-cells; cannabinoid receptor *Cnr1* which was enriched in K-cells; and *Ffar1*, *Gpr119* and *Avpr1b* which were expressed in the same region of the map as L- and N-cells. EC-cells showed notable enrichment for the sodium channel subunit *Scn3a*, and the short chain fatty acid receptor *Ffar2*, while a subset of EC-cells were enriched for *Trpa1*. A handful of genes were enriched in multiple EEC subgroups, including *Glp1r* which was co-expressed with both *Sst* (D-cells) and *Tph1* (EC-cells) (S3 Fig). The expression levels of differentially expressed GPCRs and ion channels, including the number of hormone-expressing cells in which they were detected are depicted in detail in S3 Fig.

The identification of *Mc4r* in I-cells was interesting as this receptor has previously been implicated in GLP-1 release from the distal gut, and has been identified but not functionally characterised in EECs purified from Cck-Cre mice [25, 26]. In the USI of the current dataset, *Mc4r* was expressed in 55% of the large population of *Cck*-positive cells, and 95% of *Mc4r*-expressing cells co-expressed *Cck* (S3 Fig). By contrast *Mcr4* was only co-expressed in 2% (3/ 145) of *Gcg*-positive cells in the lower SI and 7% (26/384) in the LI and in 3% of K-cells in the upper SI.

To evaluate the functional significance of the observed differential expression of *Mc4r* in *Cck*-expressing but not *Gip*-expressing cells, we assessed secretion of GIP by ELISA and of CCK using a new mass-spectrometry assay for mouse preproCCK(21–44), mirroring use of human preproCCK(21–44) as a surrogate for CCK secretion [20] (Fig 3B). Setmelanotide (BIM-22493 or RM-493), a synthetic MC4R agonist, induced a 2.1-fold increase ($p < 0.0001$) in CCK secretion from mouse upper small intestinal epithelial cultures, but did not significantly increase GIP release (S3 Fig). These results suggest that functional MC4R's are present in I-cells and play a role in the release of CCK.

### Response of individual EECs to multiple nutrients

To examine whether individual EECs are responsive to multiple stimuli, we examined whether known sensors for carbohydrates (*Sglt1*), lipids (*Ffar1*, *Ffar4*, *Gpr119*), bile acids (*Gpbar1*) and aromatic amino acids (*Gpr142*, *Casr*) are co-expressed by single EECs (Fig 4). High proportions of I-, K-, L- and N-cells expressed these nutrient sensors and many EECs expressed two or more. Expression data were validated in single cell calcium recordings from mouse ileal organoids, in which individual L-cells were observed to exhibit calcium responses to glucose, AM1638 (an FFA1 agonist) and tryptophan (Fig 4C–4E). All cells tested responded to at least two stimuli, and 60% (12 of 20) responded to all three, confirming the pluri-responsiveness of single EECs.

### Discussion

This paper provides a comprehensive transcriptomic and clustering analysis of EECs from the full length of the mouse GI tract. Whereas some hormones and cell types are regionally restricted (such as ghrelin, gastrin and histamine in the stomach), others such as EC-cells are found along the length of the gut. Combining data from 20,006 EECs provided the power to generate robust cluster maps, revealing that EECs tend to cluster with other EECs producing the same hormone or groups of hormones.

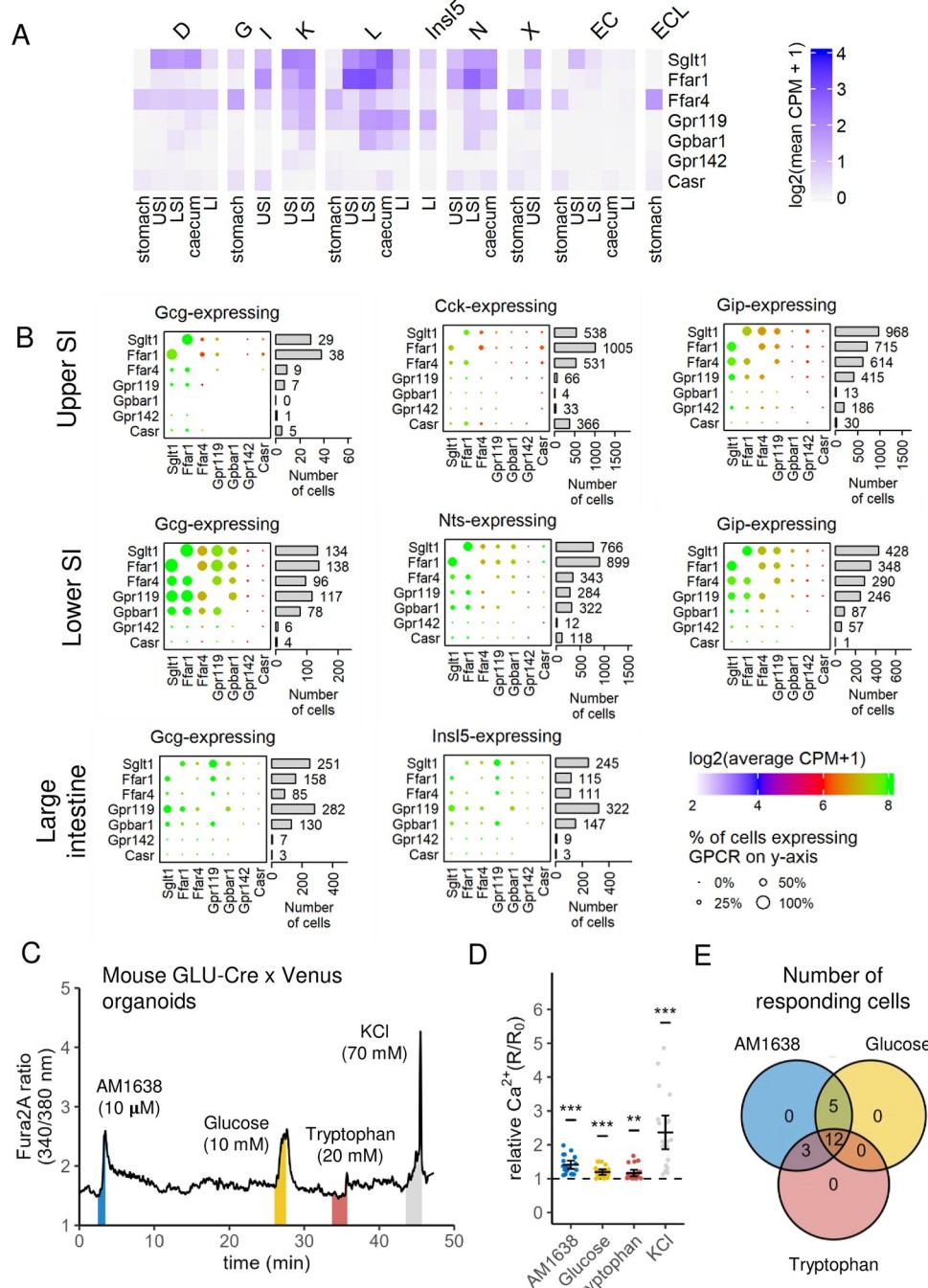

**Fig 4. EECs respond to multiple nutrients.** (A) Mean expression of selected nutrient-sensing GPCRs, *Sglt1* (carbohydrates), *Ffar1*, *Ffar4* and *Gpr119* (lipids), *Gpbar1* (bile acids), and *Gpr142* and *Casr* (aromatic amino acids), per clustered EEC type per GI region. Expression is $\log_2$(mean counts per million + 1). (B) Dot plots representing co-expression of nutrient-sensing GPCRs, in cells expressing given hormones, per GI region. Cells represented include *Gcg-*, *Cck-*, and *Gip*-expressing cells in the USI, *Gcg-*, *Nts-*, and *Gip*-expressing cells in the LSI, and *Gcg-* and *Insl5*-expressing cells in the LI. Dot size represents the percentage of cells expressing the GPCR on the y-axis, dot colour the expression of the GPCR on the x-axis in $\log_2$(mean counts per million + 1). Bars on the right represent the number of cells expressing the GPCR on the y-axis. GPCR and ion channel genes in (A) and (B) are selected as the top 20 most-expressed differentially-expressed ($p_{adj} < 0.05$) genes of that type, per GI region. (C) Example trace of $Ca^{2+}$ levels in a murine L-cell, from organoids derived from GLU-Cre x Venus mice, treated in series with AM1638 (10 µM; blue), glucose (10 mM; yellow), tryptophan (20 mM; red), and positive-control KCl (70 mM; grey), separated by wash steps. The time over which each treatment was imposed is represented by the coloured vertical bars. $Ca^{2+}$ levels were

measured using fluorescent dye Fura2-AM, and are depicted as the 340/380 ratio. (D) $Ca^{2+}$ levels (fura2 340/380 ratios) during treatment (R) relative to baseline (R0, measured immediately prior to treatment). Horizontal lines represent mean levels (n = 4 experiments, 3–6 cells per experiment). Statistical analysis by one-tailed Student's t-test ($\mu$ = 1; *** $p_{adj} < 10-3$). (E) Venn diagram of cells from (D) responding to each treatment, demonstrating ability of some cells to respond to multiple treatment types.

A number of EECs formed distinct clusters, as particularly evident for X-, D-, K-, ECL-, EC- and G-cells, with little co-expression of hormonal markers from other cell groups, except for *Sct* which was found in all EEC clusters below the stomach. Cells expressing *Cck*, *Gcg*, *Nts*, *Sct*, *Pyy* and *Insl5* overlapped substantially and individual EECs in these groups frequently expressed more than one hormone from the group, supporting the view that these cells can be considered members of one large EEC family. Individual hormonal profiles in these cells may depend on factors such as their location in the GI tract and stage of maturity. Despite the close similarity between K- and L-cells by bulk RNA sequencing [5], these cell types formed separate clusters, as also reported by other single cell RNA sequencing analyses of the small intestine [8–10].

Transcriptomic and functional analyses of EEC populations in mice and humans have identified a number of receptors involved in eliciting post-prandial gut hormone secretion, including *Ffar1-4*, *Gpbar1*, *Gpr119* and *Casr*. One of the major next challenges in the field is to determine which of these EEC-receptor partnerships are important for post-prandial physiology and which could potentially be targeted pharmacologically to increase endogenous gut hormone release for treating metabolic diseases such as type 2 diabetes and obesity. In our combined dataset most GPCRs were detectable in a range of EEC subtypes although expressed at higher levels in some than others, with *Ffar1* showing particularly high expression in the broad cluster of L-cells.

By performing cluster analysis using genes in the GPCR cell signalling space, we found that EECs continued to separate by their original-defined cell type, suggesting that the clustering of EECs is not only driven by their major DE hormones but also by their sensory and signalling machinery. This supports the view that individual clusters may be tuned towards different stimuli. The analysis supported previous reports that EC-cells lack receptors for most nutrient stimuli but have relatively high expression of receptors for SCFA (*Ffar2*) and enteric hormones [27], such as *Glp1r*, *Npy1r* and *Galr1*. Gastric X-cells, by contrast exhibit particularly high expression of *Ffar4*, a relatively poorly understood receptor that has been linked previously to stimulation of GLP-1 release but inhibition of gastric ghrelin and SST secretion [28–30]. Our analysis of the GPCR cell signalling space did not uncover an explanation for the opposing signalling downstream of FFAR4 in these cell types. Interestingly, gastric and upper small intestinal X-cells formed separate clusters in the GPCR signalling space, consistent with the idea that ghrelin release in the stomach and duodenum are regulated by different stimuli; gastric ghrelin is typically elevated in the fasting state and suppressed by feeding, whereas the regulation of duodenal ghrelin release is not well understood [31], although in humans, duodenal ghrelin is colocalised with motilin (which is not expressed in rodents), and studies in human duodenal organoids revealed stimulation of motilin release by nutritional stimuli [32].

In the GPCR cell signalling analysis, I-cells separated from L- and N-cells, despite their close overlap at the total transcriptome level. An interesting finding from this analysis was the abundance of *Cck*-expressing cells in the upper small intestine that co-expressed *Mc4r*, also recently noted in Hayashi et al. [9]. Previously, however, *Mc4r* expression had been identified in colonic L-cells [25] and linked to GLP-1 release from the distal gut [26]. In primary colonic cultures we were previously unable to detect enhanced GLP-1 release in cell supernatants following MC4R activation, although secretion was inferred by short circuit currents [25]. The

low percentage of *Gcg*-expressing cells in the colon that co-expressed *Mc4r* (7%) likely explains these difficulties in measuring MC4R-dependent GLP-1 release in mixed colonic cultures. By contrast, CCK release from upper small intestinal cultures was strongly stimulated by α-MSH and setmelanotide, suggesting that I-cells may be a previously unrecognised target for anorexic POMC-derived peptides.

With the power of analysing many cells, we were also able to confirm that many individual EECs express sensory machinery to detect multiple stimuli, as confirmed by calcium imaging data. This supports previous ideas that EECs are pluri-responsive and not individually tuned to respond to specific stimuli.

## Conclusions

This comprehensive transcriptomic map of the mouse EEC system should be a valuable tool for understanding which EEC subtypes are most likely to respond to specific pharmacological stimuli, and the impact of cell location along the GI tract. The data are openly accessible, and available for researchers to re-analyse and interrogate for receptors or other pathways of interest. The findings from analysing EECs in the GPCR cell signalling space raise the possibility of performing similar analyses using other gene sets of interest such as transcription factors, to enable deeper understanding of this diverse EEC population.

Previous studies have shown relatively good translatability to humans of findings derived from study of the mouse enteroendocrine system, with most EEC GPCRs identified in murine studies having similar activity in human EECs. Developing small molecular agonists to increase secretion from EECs in humans is a strategy of translational interest, as reproducing the dramatic elevations of gut hormones such as GLP-1 and PYY observed after bariatric surgery is predicted to mimic some of the beneficial effects of surgery on appetite suppression and insulin release.

## Supporting information

**S1 Fig. Clustering supporting details and integration with unbiased regional datasets.** (A) Numbers of cells in each cluster from Fig 1A. (B) Expression of *Pcsk1* and *Pcsk2* in each EEC clusters per GI region. Expression is $\log_2$(mean counts per million + 1). (C) tSNE plot for integrated dataset of all EECs from all GI regions. Left: Samples labelled by GI region. Right: Samples labelled by GI region for D, K, L, N, X, and ECs only. (D,E) Data from the small and large intestine were separately integrated with previously published single cell RNA sequencing datasets, including (D) mouse small intestine epithelial cells (7) and (E) wild-type mouse colon cells [18]. tSNE maps show overlap in EECs from the published (red) and our dataset (blue), amongst non-EECs (grey) from each dataset.
(TIF)

**S2 Fig. EEC marker raw count histograms.** Histogram of raw counts per EEC marker (left to right) per GI region (top to bottom). Huang threshold labelled by red vertical line.
(TIF)

**S3 Fig. GPCR and ion channel expression across EECs by region.** Dot plots representing co-expression of hormones (or *Tph1*) and (A) GPCRs, (B) ion channels, and (C) enteric hormone receptors, per GI region. Dot size represents the percentage of cells expressing the hormone on the y-axis, dot colour the expression of the gene on the x-axis in $\log_2$(mean counts per million + 1). Bars on the right represent the number of cells expressing the hormone on the y-axis. GPCR and ion channel genes in (A) and (B) are selected as the top 20 most-expressed differentially-expressed ($p_{adj} < 0.05$) genes of that type, per GI region. (B) Total GIP secretion,

measured by ELISA, from mouse USI primary tissue after 2 hr stimulation with setmelanotide (10 μM), with DMSO (2%), and IBMX (10 μM)/forskolin (10 μM)(FI) as negative and positive controls, respectively. Concentrations in test supernatants are shown normalised to mean concentrations in DMSO controls, per experiment. Horizontal lines represent the mean (n = 3 experiments, 3 samples per experiment). Statistical analysis by two-way ANOVA, and Tukey's HSD post-hoc test (*** p.adj $< 10^{-3}$).
(TIF)

## Acknowledgments

We thank the MRL Genomics and Transcriptomics Core, Peptidomics Core and Disease Model Core; the Flow Cytometry Core at Cambridge Institute for Medical Research, and Genomics Cores at CRUK Cambridge Institute and Stem Cell Institute. We thank Andrew Leiter for the original generation and provision of NeuroD1-Cre mice.

For the purpose of open access, the author has applied a Creative Commons Attributions (CC BY) license to any Author Accepted Manuscript version arising from this submission.

## Author Contributions

**Conceptualization:** Frank Reimann, Fiona M. Gribble.

**Data curation:** Christopher A. Smith.

**Formal analysis:** Christopher A. Smith.

**Funding acquisition:** Frank Reimann, Fiona M. Gribble.

**Investigation:** Christopher A. Smith, Elisabeth A. A. O'Flaherty, Nunzio Guccio, Austin Punnoose, Tamana Darwish, Jo E. Lewis, Rachel E. Foreman, Alice E. Adriaenssens, Fiona M. Gribble.

**Methodology:** Christopher A. Smith, Austin Punnoose, Jo E. Lewis, Rachel E. Foreman, Richard G. Kay, Alice E. Adriaenssens, Frank Reimann, Fiona M. Gribble.

**Project administration:** Frank Reimann, Fiona M. Gribble.

**Resources:** Joyce Li, Frank Reimann, Fiona M. Gribble.

**Software:** Christopher A. Smith.

**Supervision:** Richard G. Kay, Alice E. Adriaenssens, Frank Reimann, Fiona M. Gribble.

**Validation:** Christopher A. Smith.

**Visualization:** Christopher A. Smith.

**Writing – original draft:** Christopher A. Smith, Frank Reimann, Fiona M. Gribble.

**Writing – review & editing:** Christopher A. Smith, Elisabeth A. A. O'Flaherty, Nunzio Guccio, Tamana Darwish, Jo E. Lewis, Rachel E. Foreman, Joyce Li, Richard G. Kay, Alice E. Adriaenssens, Frank Reimann, Fiona M. Gribble.

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
