## [Decision Letter · Decision Letter 0]

22 Feb 2024

PONE-D-24-02587Single-cell transcriptomic atlas of enteroendocrine cells along the murine gastrointestinal tractPLOS ONE

Dear Dr. Gribble,

Thank you for submitting your manuscript to PLOS ONE. After careful consideration, we feel that it has merit but does not fully meet PLOS ONE’s publication criteria as it currently stands. Therefore, we invite you to submit a revised version of the manuscript that addresses the points raised during the review process.

We look forward to receiving your revised manuscript.

Kind regards,

Mohammad Sadegh Taghizadeh, Ph.D.

Academic Editor

PLOS ONE

Journal Requirements:

“Research in the Reimann/Gribble laboratories was funded by Wellcome (220271/Z/20/Z) and MRC (MRC_MC_UU_12012/3). The MS instrument was funded by the MRC “Enhancing UK clinical research” grant (MR/M009041/1). MRL Genomics and Transcriptomics Core, Disease Model Core and Peptidomics Core were supported by funding from MRC (MRC_MC_UU_12012/5).”

“FMG and FR have received funding for other projects from AstraZeneca and Eli Lilly.”

We note that one or more of the authors are employed by a commercial company: AstraZeneca and Eli Lilly

Reviewers' comments:

Reviewer's Responses to Questions

**Comments to the Author**

1. Is the manuscript technically sound, and do the data support the conclusions?

Reviewer #1: Yes

Reviewer #2: Partly

Reviewer #3: Yes

2. Has the statistical analysis been performed appropriately and rigorously? 

Reviewer #1: Yes

Reviewer #2: Yes

Reviewer #3: Yes

3. Have the authors made all data underlying the findings in their manuscript fully available?

Reviewer #1: Yes

Reviewer #2: Yes

Reviewer #3: Yes

4. Is the manuscript presented in an intelligible fashion and written in standard English?

Reviewer #1: Yes

Reviewer #2: Yes

Reviewer #3: Yes

5. Review Comments to the Author

Reviewer #1: 

Enteroendocrine cells (EECs) are rare hormone-producing cells in the intestinal epithelium. Distinct subtypes of EECs produce unique combinations of hormones, which regulate metabolism, bowel movement and appetite. Due to their scarcity, defining EEC heterogeneity at the transcriptome level has been challenging. In this study, the authors utilize a genetically encoded EEC reporter to generate a scRNAseq atlas comprising ~20K EECs from the proximal to distal mouse gastrointestinal tract. They describe regional variations in EEC subtype abundance, and showcase several GPCRs with a biased expression in specific EEC subtypes. As functional validation, they demonstrate that a specific GPCR agonist induces EEC activation in vitro. Overall, this study provides a comprehensive resource for the EEC field. Due to the existence of multiple single cell RNA atlases in the mouse gut that cover many EECs, this advance is however rather incremental. Please find below my point-by-point suggestions.

- In figure 3: It is unclear to what extent reclustering based on GPCR expression allowed the authors to discover differences in GPCR distribution across EEC subtypes. It seems likely that differential expression on the initial clustering would have revealed the same.

- Figure 4: The authors should include measurements of other hormones in response to Mc4r agonists, to stress the Cck-specific signaling.

- The authors succinctly write in lines 418/419 that the identification of Mc4r in I cells is interesting, as previous work showed its role in GLP-1 release. Reference 24 does indeed show this/L cell expression, but also evidenced I cell expression using a CCK-reporter mouse. The authors should therefore mention Mc4r was previously found to be enriched in I cells.

- It appears that in figure 4B the legend erroneously indicates that the dot size represents the percentage of cells expressing a certain hormone, instead of a GPCR.

- Is there any bias in isolating certain EEC subtypes using the NeuroD1 tracer? All EEC subsets seem to be present, but a statement/reference on this would be helpful. Stomach endocrine cell distribution appears different than reported (should be more biased for EC/ECL). In addition: could the authors comment on the distribution of ECL cells across GI tract regions. It is present in equal numbers across the GI tract, while exclusive presence in the stomach is expected. This is also stated in line 294 (the authors similarly state this is absolute for G cells, which again is found in other regions).

- I could not find any statement on the average number of reads/genes detected per cell. This would be important, including a comparison to previous murine gut sequencing studies. As much as the number of cells sequenced, the depth per cell is key to identify lowly expressed genes such as GPCRs.

Reviewer #2: 

This manuscript from an eminent laboratory on enteroendocrine cell (EEC) biology compares mouse stomach, proximal and distal small intestine, caecum, and colonic EEC transcriptomes at single-cell resolution. Knowing that EEC physiology is tied to G protein-coupled receptor (GPCR) signaling, the authors highlight differences and overlaps in GPCR mRNA expression across EECs and GI regions. They show that a range of nutrients trigger Ca++ flux in ileal L cells differentiated in vitro.

Several scRNA-seq profiles of EECs have been reported, including some by this group, but this is the first to consider the full length of the mouse GI tract together (colonic EEC data are taken from a previous study that is cited). The work is performed to the high standards of the senior authors, who are recognized experts in the field. The quality of the data seems high and the story is presented clearly. Although certain conclusions, such as that most EECs express more than one bioactive peptide, have been reported previously, these aggregate data from >20,000 cells will make a useful contribution to the literature. Demonstration of cholecystokinin (CCK) release in response to multiple stimulants (Fig. 3B) adds one physiologic dimension to the study.

My only question pertains to the second physiologic dimension: Ca++ flux that various agonists trigger in Venus-labeled L cells isolated from differentiated ileal organoids (Fig. 4C-E). The range of values shown on the graph (Fig. 4D) raising questions about what is real signal and what is background. The implication of the dashed line on the graph is that any R/R0 value >1.0 is signal and therefore legitimate to represent in the Venn diagram (Fig. 4E) as a “reactive” cell. However, the example trace in Fig. 4C suggests that many of the values clustered near 1.0 may represent background noise. Only some of those cells may merit inclusion within the Venn representation. The authors should represent the data with clear and conservative definitions of real signals or establish that even small deviations over 1.0 are real. Data from a larger number of cells may also add clarity and rigor.

Reviewer #3: 

The study entitled “Single-cell transcriptomic atlas of enteroendocrine cells along the murine gastrointestinal tract” by Smith et al. explores enteroendocrine cells (EECs) diversity and expression along the gastrointestinal tract by single-cell RNA-sequencing of GFP-labeled sorted EECs from different tissues and locations along the tract (from the stomach to rectum). The authors identified ten main EEC clusters within the different tissues with similar transcriptional profiles, showing multiple expressions of a few gut hormones in specific subsets. The authors investigated GPCRs of the different EEC types and showed that GPCR expression correlates with the gene expression of the cells. The authors propose to utilize this approach to test other specific factors within the cells, such as transcription factors, to investigate the diverse EEC population and properties.

My only comment is the disadvantage of the EEC isolation method that relies on a GFP reporter under the specific TF associated with most EECs, NeuroD1. Using this method allows the enrichment of EECs from EpCAM+ cells, which are about 2-4% of total EpCAM, but could miss EEC subsets that are NeuroD1 negative. It would be interesting to compare the data obtained from this study to an unbiased analysis of EECs extracted from the total epithelial cells of one tissue.

In conclusion, the authors performed elegant experiments to characterize the EEC subsets of different gastrointestinal compartments and test for hormones and GPCRs associated with them.

This manuscript will surely provide valuable data and guidelines for further research in EEC, food intake, and metabolic disorders in general.

6. PLOS authors have the option to publish the peer review history of their article (what does this mean?). If published, this will include your full peer review and any attached files.

Reviewer #1: No

Reviewer #2: No

Reviewer #3: No

---

## [Author Response · Author response to Decision Letter 0]

4 Jun 2024

Reviewer #1: 

Enteroendocrine cells (EECs) are rare hormone-producing cells in the intestinal epithelium. Distinct subtypes of EECs produce unique combinations of hormones, which regulate metabolism, bowel movement and appetite. Due to their scarcity, defining EEC heterogeneity at the transcriptome level has been challenging. In this study, the authors utilize a genetically encoded EEC reporter to generate a scRNAseq atlas comprising ~20K EECs from the proximal to distal mouse gastrointestinal tract. They describe regional variations in EEC subtype abundance, and showcase several GPCRs with a biased expression in specific EEC subtypes. As functional validation, they demonstrate that a specific GPCR agonist induces EEC activation in vitro. Overall, this study provides a comprehensive resource for the EEC field. Due to the existence of multiple single cell RNA atlases in the mouse gut that cover many EECs, this advance is however rather incremental. Please find below my point-by-point suggestions.

- In figure 3: It is unclear to what extent reclustering based on GPCR expression allowed the authors to discover differences in GPCR distribution across EEC subtypes. It seems likely that differential expression on the initial clustering would have revealed the same.

Response: We agree with the reviewer that there are different ways in which the GPCR distribution can be analysed across different EEC populations. In S3 Fig we took a traditional approach and analysed individual GPCR expression across different EEC populations defined by their expression of a hormone of interest. As expected, this revealed differential GPCR across EEC populations. The aim of the experiment in figure 3 was to test the reverse hypothesis, that the EEC populations differ by how they signal. We believe this is an interesting alternative approach to compare cell populations, and the finding of differential Mc4r expression in CCK-cells was one of the key outcomes. 

- Figure 4: The authors should include measurements of other hormones in response to Mc4r agonists, to stress the Cck-specific signaling.

Response: We thank the reviewer for making this suggestion and have performed additional experiments to test the hypothesis that Mc4r agonism would not affect secretion of GIP. As predicted, setmelanotide increased secretion of CCK but not of GIP from primary mouse upper small intestinal epithelial cultures. The additional CCK results have been added to the original data, increasing the n-number, and the GIP results are shown in a new subfigure (S3 Fig). As these were upper small intestinal cultures, GLP-1 and PYY levels were too low to quantify, so we restricted the new analysis to CCK and GIP.

- The authors succinctly write in lines 418/419 that the identification of Mc4r in I cells is interesting, as previous work showed its role in GLP-1 release. Reference 24 does indeed show this/L cell expression, but also evidenced I cell expression using a CCK-reporter mouse. The authors should therefore mention Mc4r was previously found to be enriched in I cells.

Response: We thank the reviewer for noticing that we had not referred to the CCK-cell data in this reference. This is now included, and the text reads:

“The identification of Mc4r in I-cells was interesting as this receptor has previously been implicated in GLP-1 release from the distal gut, and has been identified but not functionally characterised in EECs purified from Cck-Cre mice (25, 26)” 

- It appears that in figure 4B the legend erroneously indicates that the dot size represents the percentage of cells expressing a certain hormone, instead of a GPCR.

Response: Thank you, this is now corrected.

- Is there any bias in isolating certain EEC subtypes using the NeuroD1 tracer? All EEC subsets seem to be present, but a statement/reference on this would be helpful. Stomach endocrine cell distribution appears different than reported (should be more biased for EC/ECL). In addition: could the authors comment on the distribution of ECL cells across GI tract regions. It is present in equal numbers across the GI tract, while exclusive presence in the stomach is expected. This is also stated in line 294 (the authors similarly state this is absolute for G cells, which again is found in other regions).

Response: As also questioned by reviewer 2, we have now systematically examined whether the NeuroD1-Cre reporter labelled EECs in an unbiassed manner. As shown in the new S1 Fig (D, E), we combined our new scRNAseq data with published scRNAseq data from the epithelium of the upper SI (Haber et al 2017) and large intestine (Xu et al 2023). Our EEC populations showed good overlap with the (relatively small) EEC populations in these unbiassed datasets, and we did not identify any EEC populations in the unbiassed data that had not been detected using the NeuroD1 marker. The revised version contains additional methods, and the following text in the results: 

“We combined the data from our USI and LSI NeuroD1-labelled EECs with published unbiassed datasets (7, 18). EECs from the current study showed good overlap with EEC populations in the unbiassed datasets (S1 Fig), and no additional EEC populations were evident in the full epithelial data that had not been picked up using the NeuroD1-Cre approach.”

We thank the reviewer for noticing that the Huang thresholding method (shown in S2 Fig as the method used to define positive cells in Fig 2), calculated that there are more ECL cells than expected in regions beyond the stomach. This is because the method looks at the range of expression of the marker gene across all the cells in the region, and calls those with the highest expression as “positive” even if the expression level is very low. Use of a thresholding method is necessary because transcripts for hormones that are highly expressed in a tissue are detectable in the majority of negative cells, likely because high abundance RNA released from a few burst cells can be picked up in all droplets during the 10X analysis. We prefer not to tweak the Huang thresholding programme manually to generate an expected outcome, so have instead added extra colour coded circles in figure 2 to show average expression of the marker gene in its positive cell population in each region. This shows up as a series of circles across the diagonal, with the colour varying by region e.g. Hdc is now seen to be more highly expressed in the Hdc-positive cells in the stomach than in Hdc-positive cells in the colon/rectum (where it is co-expressed at low levels in L-cells). We hope this provides some additional clarity, and have added the following text to explain the data:

“Use of the Huang thresholding method identified similar numbers of EECs as the clustering method for most cell types, but over-estimated ECL cell numbers in regions beyond the stomach where low Hdc expression in other EEC types was called positive by the algorithm (Fig 2 A). “ 

We have also now presented the cell numbers generated from the clustering in fig1A, as an alternative estimate of cell cluster sizes in different regions of the gut (S1 Fig). This again reveals that the Huang thresholding method has worked well for most cell types, but has broken down for ECL cells in regions where both cell abundance and Hdc expression are low. 

- I could not find any statement on the average number of reads/genes detected per cell. This would be important, including a comparison to previous murine gut sequencing studies. As much as the number of cells sequenced, the depth per cell is key to identify lowly expressed genes such as GPCRs.

Response: Apologies for the omission of this important piece of information. The following text has now been added to the methods:

“Of the filtered data, the median (min – max) number of reads per cell were 35,679 (2,140 – 389,084) in the stomach, 15,440 (718 – 261,381) in the USI, 6,462 (650 – 216,128) in the LSI, 5,664 (1,648 – 104,978) in the caecum, and 4,438 (1,402 – 40,537) in the LI dataset. The median (min – max) number of genes per cell were 3,276 (1,002 – 9,330) in the stomach, 3,663 (504 – 9,350) in the USI, 2,487 (501 – 9,367) in the LSI, 2,043 (1,001 – 8,308) in the caecum, and 1,943 (1,001 – 5,935) in the LI dataset.”

Reviewer #2: 

This manuscript from an eminent laboratory on enteroendocrine cell (EEC) biology compares mouse stomach, proximal and distal small intestine, caecum, and colonic EEC transcriptomes at single-cell resolution. Knowing that EEC physiology is tied to G protein-coupled receptor (GPCR) signaling, the authors highlight differences and overlaps in GPCR mRNA expression across EECs and GI regions. They show that a range of nutrients trigger Ca++ flux in ileal L cells differentiated in vitro.

Several scRNA-seq profiles of EECs have been reported, including some by this group, but this is the first to consider the full length of the mouse GI tract together (colonic EEC data are taken from a previous study that is cited). The work is performed to the high standards of the senior authors, who are recognized experts in the field. The quality of the data seems high and the story is presented clearly. Although certain conclusions, such as that most EECs express more than one bioactive peptide, have been reported previously, these aggregate data from >20,000 cells will make a useful contribution to the literature. Demonstration of cholecystokinin (CCK) release in response to multiple stimulants (Fig. 3B) adds one physiologic dimension to the study.

My only question pertains to the second physiologic dimension: Ca++ flux that various agonists trigger in Venus-labeled L cells isolated from differentiated ileal organoids (Fig. 4C-E). The range of values shown on the graph (Fig. 4D) raising questions about what is real signal and what is background. The implication of the dashed line on the graph is that any R/R0 value >1.0 is signal and therefore legitimate to represent in the Venn diagram (Fig. 4E) as a “reactive” cell. However, the example trace in Fig. 4C suggests that many of the values clustered near 1.0 may represent background noise. Only some of those cells may merit inclusion within the Venn representation. The authors should represent the data with clear and conservative definitions of real signals or establish that even small deviations over 1.0 are real. Data from a larger number of cells may also add clarity and rigor.

Response: We thank the reviewer for noticing that we had not included a description of how we defined a cell as a “responder”. The cut-off for defining responders was set at a relatively conservative increase by 10% in the fura-2 ratio during addition of the test compound. Based on extensive previous experience, this cut-off is more likely to generate false negatives than false positives. This has been added to the methods, which now reads:

“Cells exhibiting increases in the fura2 fluorescence ratio of >10% during perfusion of the test agent were counted as responders – a definition that is more likely to generate false negatives than false positives.” 

Reviewer #3: 

The study entitled “Single-cell transcriptomic atlas of enteroendocrine cells along the murine gastrointestinal tract” by Smith et al. explores enteroendocrine cells (EECs) diversity and expression along the gastrointestinal tract by single-cell RNA-sequencing of GFP-labeled sorted EECs from different tissues and locations along the tract (from the stomach to rectum). The authors identified ten main EEC clusters within the different tissues with similar transcriptional profiles, showing multiple expressions of a few gut hormones in specific subsets. The authors investigated GPCRs of the different EEC types and showed that GPCR expression correlates with the gene expression of the cells. The authors propose to utilize this approach to test other specific factors within the cells, such as transcription factors, to investigate the diverse EEC population and properties.

My only comment is the disadvantage of the EEC isolation method that relies on a GFP reporter under the specific TF associated with most EECs, NeuroD1. Using this method allows the enrichment of EECs from EpCAM+ cells, which are about 2-4% of total EpCAM, but could miss EEC subsets that are NeuroD1 negative. It would be interesting to compare the data obtained from this study to an unbiased analysis of EECs extracted from the total epithelial cells of one tissue.

Response: We thank the reviewer for this comment, and have now integrated our data with the single cell data from upper small intestine (Haber 2017) and large intestine (Xu 2023). The merged clustering analysis is included in a new S1 Fig D,E. The analyses showed good overlap between the EECs from the different studies, but did not identify any EEC populations in the unbiassed datasets that were not picked up using NeuroD1-Cre as a reporter. We conclude that our analysis did successfully identify all the different EEC populations derived from the gut epithelium in the small intestine, and have added this to the methods and results. The new text in the results is:

“We combined the data from our USI and LSI NeuroD1-labelled EECs with published unbiassed datasets (7, 18). EECs from the current study showed good overlap with EEC populations in the unbiassed datasets (S1 Fig), and no additional EEC populations were evident in the full epithelial data that had not been picked up using the NeuroD1-Cre approach.”

In conclusion, the authors performed elegant experiments to characterize the EEC subsets of different gastrointestinal compartments and test for hormones and GPCRs associated with them.

This manuscript will surely provide valuable data and guidelines for further research in EEC, food intake, and metabolic disorders in general.

Response: Thank you for the positive comments

---

## [Decision Letter · Decision Letter 1]

21 Jun 2024

PONE-D-24-02587R1Single-cell transcriptomic atlas of enteroendocrine cells along the murine gastrointestinal tractPLOS ONE

Dear Dr. Gribble,

Thank you for submitting your manuscript to PLOS ONE. After careful consideration, we feel that it has merit but does not fully meet PLOS ONE’s publication criteria as it currently stands. Therefore, we invite you to submit a revised version of the manuscript that addresses the points raised during the review process.

We look forward to receiving your revised manuscript.

Kind regards,

Mohammad Sadegh Taghizadeh, Ph.D.

Academic Editor

PLOS ONE

Journal Requirements:

Reviewers' comments:

Reviewer's Responses to Questions

**Comments to the Author**

1. If the authors have adequately addressed your comments raised in a previous round of review and you feel that this manuscript is now acceptable for publication, you may indicate that here to bypass the “Comments to the Author” section, enter your conflict of interest statement in the “Confidential to Editor” section, and submit your "Accept" recommendation.

Reviewer #1: All comments have been addressed

Reviewer #2: (No Response)

2. Is the manuscript technically sound, and do the data support the conclusions?

Reviewer #1: Yes

Reviewer #2: Yes

3. Has the statistical analysis been performed appropriately and rigorously? 

Reviewer #1: Yes

Reviewer #2: I Don't Know

4. Have the authors made all data underlying the findings in their manuscript fully available?

Reviewer #1: Yes

Reviewer #2: Yes

5. Is the manuscript presented in an intelligible fashion and written in standard English?

Reviewer #1: Yes

Reviewer #2: Yes

6. Review Comments to the Author

Reviewer #1: 

The authors have addressed my questions and I recommend publication in Plos One. I am convinced this will be a very useful resource to the community.

A remaining note on the cell clustering in Figure 2:

The authors comment that the Huang thresholding calls cell types based on transcript levels are relative to cell in that particular GI tract region. It may be better to combine all regions for the tresholding and then separately present the data? To me it is still somewhat confusing to present these high HDC+ numbers in the distal gut. This may also resolve some other 'stretched' annotations (e.g., now it appears there are relatively equal number of Gcg+ cells in the proximal and distal gut, INSL5 in the proximal gut..). More GIP than GCG in the distal gut? This is perhaps more a point of preference of how to present this, and I appreciate the addition of average gene expression as a clarification.

Reviewer #2: 

I was the most positive of the 3 referees, with significant concern only about the Fura2A ratios represented in Fig. 4C-E. The authors’ response is that a 10% increase in the ratio is “relatively conservative” and that “based on extensive previous experience,” that “cut-off is more likely to generate false negatives than false positives.” This response and insertion of a sentence to that effect in the Methods are less than satisfactory. Unless I don’t understand the tracing (Fig. 4C) or how values are graphed in Fig. 4D, the distribution of values within any group seems to vary by more than 10% and the responses to Glucose or Tryptophan look very different in Fig. 4C. I don’t challenge the authors’ conservatism, nor do I need to stand in the way of publication, but other readers may share my concern (confusion?) and deserve more clarity about the assay and its signal/noise parameters. I otherwise like this paper and believe it makes a useful contribution to the field.

7. PLOS authors have the option to publish the peer review history of their article (what does this mean?). If published, this will include your full peer review and any attached files.

Reviewer #1: No

Reviewer #2: No

---

## [Author Response · Author response to Decision Letter 1]

17 Jul 2024

Reviewer #1: 

A remaining note on the cell clustering in Figure 2:

The authors comment that the Huang thresholding calls cell types based on transcript levels are relative to cell in that particular GI tract region. It may be better to combine all regions for the tresholding and then separately present the data? To me it is still somewhat confusing to present these high HDC+ numbers in the distal gut. This may also resolve some other 'stretched' annotations (e.g., now it appears there are relatively equal number of Gcg+ cells in the proximal and distal gut, INSL5 in the proximal gut..). More GIP than GCG in the distal gut? This is perhaps more a point of preference of how to present this, and I appreciate the addition of average gene expression as a clarification.

Response: We thank the reviewer for this question, which triggered us to look yet again at the Huang thresholding method. We tried several ways of combining the data, and have now made 2 adjustments to the method: (1) we have combined data from all regions, as the reviewer suggested, with the exception of Ghrl and Sst in the stomach which were so highly expressed that they resulted in background levels in other stomach EECs that were higher than in Ghrl and Sst-expressing cells in the small intestine – this was a major reason we had originally treated each region independently; (2) we have used CPM instead of raw counts when combining the data. The new calculations still identify Hdc-expressing cells in the distal gut, but these are not a distinct cell type, rather there is some Hdc expression in other EECs, particularly enterochromaffin cells and L-cells. We have changed the description of this result to make it clear that this is not a distinct ECL population. The new text reads:

“Low levels of Hdc expression were unexpectedly identified in some EECs in the distal intestine, predominantly colocalised in EC cells in the caecum and with L-cell hormones in the large intestine.”

Reviewer #2: 

I was the most positive of the 3 referees, with significant concern only about the Fura2A ratios represented in Fig. 4C-E. The authors’ response is that a 10% increase in the ratio is “relatively conservative” and that “based on extensive previous experience,” that “cut-off is more likely to generate false negatives than false positives.” This response and insertion of a sentence to that effect in the Methods are less than satisfactory. Unless I don’t understand the tracing (Fig. 4C) or how values are graphed in Fig. 4D, the distribution of values within any group seems to vary by more than 10% and the responses to Glucose or Tryptophan look very different in Fig. 4C. I don’t challenge the authors’ conservatism, nor do I need to stand in the way of publication, but other readers may share my concern (confusion?) and deserve more clarity about the assay and its signal/noise parameters. I otherwise like this paper and believe it makes a useful contribution to the field.

Response: There is no perfect way of distinguishing a responding cell from a non-responder, since some responses can be subtle and background noise in the traces varies from cell to cell. Attempting to address the reviewer’s concern, we have taken a different approach in this revision, taking 10 readings of the fura2 ratio from just before addition of the test reagent, and 10 centred around the maximum response during perfusion with the test compound, for each cell. We have then applied a t-test between these pairs of readings and called a cell as a responder if there was a significant increase (with p<0.01). I’m not sure this is particularly superior to the 10% method, although it does call some additional cells as responders. I don’t think the exact numbers in each set of the Venn diagram are critical, as the major question was whether the cells are capable of responding to more than one stimulus type. Overall the conclusion of this section is unchanged by the method of analysis: that EECs are multi-responsive to different types of nutrient signal.

---

## [Decision Letter · Decision Letter 2]

2 Aug 2024

Single-cell transcriptomic atlas of enteroendocrine cells along the murine gastrointestinal tract

PONE-D-24-02587R2

Dear Dr. Gribble,

We’re pleased to inform you that your manuscript has been judged scientifically suitable for publication and will be formally accepted for publication once it meets all outstanding technical requirements.

Kind regards,

Mohammad Sadegh Taghizadeh, Ph.D.

Academic Editor

PLOS ONE

Additional Editor Comments (optional):

Reviewers' comments:

Reviewer's Responses to Questions

**Comments to the Author**

1. If the authors have adequately addressed your comments raised in a previous round of review and you feel that this manuscript is now acceptable for publication, you may indicate that here to bypass the “Comments to the Author” section, enter your conflict of interest statement in the “Confidential to Editor” section, and submit your "Accept" recommendation.

Reviewer #2: All comments have been addressed

2. Is the manuscript technically sound, and do the data support the conclusions?

Reviewer #2: (No Response)

3. Has the statistical analysis been performed appropriately and rigorously? 

Reviewer #2: (No Response)

4. Have the authors made all data underlying the findings in their manuscript fully available?

Reviewer #2: (No Response)

5. Is the manuscript presented in an intelligible fashion and written in standard English?

Reviewer #2: (No Response)

6. Review Comments to the Author

Reviewer #2: (No Response)

7. PLOS authors have the option to publish the peer review history of their article (what does this mean?). If published, this will include your full peer review and any attached files.

Reviewer #2: No

---

## [Editor Report · Acceptance letter]

14 Aug 2024

PONE-D-24-02587R2 

PLOS ONE

Dear Dr. Gribble, 

I'm pleased to inform you that your manuscript has been deemed suitable for publication in PLOS ONE. Congratulations! Your manuscript is now being handed over to our production team.

Kind regards, 

on behalf of

Dr. Mohammad Sadegh Taghizadeh 

Academic Editor

PLOS ONE